# A pragmatic randomized controlled trial reports lack of efficacy of hydroxychloroquine on coronavirus disease 2019 viral kinetics

Magnus Nakrem Lyngbakken [1,2], Jan-Erik Berdal[2,3], Arne Eskesen[3], Dag Kvale [2,4], Inge Christoffer Olsen[5], Corina Silvia Rueegg [5,6], Anbjørg Rangberg[7], Christine Monceyron Jonassen[7], Torbjørn Omland[1,2], Helge Røsjø [2,8 ✉] & Olav Dalgard[2,3]

Here, we randomized 53 patients hospitalized with coronavirus disease 2019 (COVID-19) to hydroxychloroquine therapy (at a dose of 400 mg twice daily for seven days) in addition to standard care or standard care alone (ClinicalTrials.gov Identifier, NCT04316377). All severe acute respiratory syndrome coronavirus 2 (SARS-CoV-2) positive patients 18 years of age or older were eligible for study inclusion if they had moderately severe COVID-19 at admission. Treatment with hydroxychloroquine did not result in a significantly greater rate of decline in SARS-CoV-2 oropharyngeal viral load compared to standard care alone during the first five days. Our results suggest no important antiviral effect of hydroxychloroquine in humans infected with SARS-CoV-2.

[1] Division of Medicine, Akershus University Hospital, Lørenskog, Norway. [2] Institute of Clinical Medicine, Faculty of Medicine, University of Oslo, Oslo, Norway. [3] Department of Infectious Diseases, Division of Medicine, Akershus University Hospital, Lørenskog, Norway. [4] Department of Infectious Diseases, Oslo University Hospital, Oslo, Norway. [5] Department of Research Support for Clinical Trials, Oslo University Hospital, Oslo, Norway. [6] Oslo Centre for Biostatistics and Epidemiology, Oslo University Hospital, Oslo, Norway. [7] Center for Laboratory Medicine, Østfold Hospital Trust, Grålum, Norway. [8] Division of Research and Innovation, Akershus University Hospital, Lørenskog, Norway. ✉email: helge.rosjo@medisin.uio.no

Hydroxychloroquine is a registered therapeutic against malaria and several autoimmune conditions, and has in vitro inhibitory effects on severe acute respiratory syndrome coronavirus 2 (SARS-CoV-2) in nontoxic concentrations[1]. In a recent report, hydroxychloroquine given more than 2 weeks after first symptoms had no effect on the clearance of SARS-CoV-2 in various respiratory tract specimens of afebrile patients hospitalized with coronavirus 2019 (COVID-19) partly pretreated with antiviral drugs[2]. Viral clearance is however an incomplete characterization of viral kinetics, and the impact of hydroxychloroquine therapy on SARS-CoV-2 viral kinetics in subjects hospitalized with COVID-19 remains to be elucidated. Hydroxychloroquine is postulated to affect viral replication, and it is reasonable to assume that an effective antiviral drug will affect respiratory tract viral titers and thereby improve the symptoms and host inflammatory responses, including the cytokine and chemokine expression that is likely responsible for many of the clinical symptoms of COVID-19[3]. Here we report findings from a two-arm, open label, pragmatic randomized controlled trial, the Norwegian Coronavirus Disease 2019 (NO COVID-19) Study (NCT04316377), that assessed the efficacy and safety of hydroxychloroquine therapy on SARS-CoV-2 oropharyngeal viral kinetics in patients hospitalized with moderately severe COVID-19. Treatment with hydroxychloroquine in addition to standard care did not result in a significantly greater rate of decline in SARS-CoV-2 oropharyngeal viral load compared to standard care alone during the first 5 days of hospitalization. Our findings suggest no important antiviral effect of hydroxychloroquine in humans infected with SARS-CoV-2.

## Results

**Patient characteristics.** From March 25, 2020, through May 25, 2020, 27 patients were randomized to hydroxychloroquine sulfate in addition to standard care and 26 patients to standard care alone (Fig. 1). Details regarding baseline demographics, clinical variables on hospital admission and safety during the trial can be found in Table 1. Time from onset of symptoms to randomization was 8 (interquartile range [IQR] 7 to 12) days in our study.

Median age was 62 (IQR 50 to 73) years, 35 patients (66.0%) were male and 2 patients (3.8%) were current smokers. Median body temperature was 38.2 (IQR 37.5 to 38.7) °C and 20 (37.7%) patients required supplemental oxygen on admission. We found no substantial differences in numbers and proportion of adverse events of special interest, serious adverse events or suspected unexpected serious adverse reactions between the hydroxychloroquine plus standard care group versus standard care alone.

**Primary outcome.** Fifty-one participants were included in the intention-to-treat analysis; 117 samples of the 133 RT-qPCR-results were above the limit of detection (2.11 $\log_{10}$ RNA copies/mL). The rate of reduction in SARS-CoV-2 viral load was 0.24 (95% CI 0.03 to 0.46) $\log_{10}$ RNA copies/mL/24 h in the hydroxychloroquine group and 0.14 (95% CI −0.10 to 0.37) $\log_{10}$ RNA copies/mL/24 h in the standard care group (reduction rate difference between the groups 0.11 [95% CI −0.21 to 0.43] $\log_{10}$ RNA copies/mL/24 h; Fig. 2). Individual RT-qPCR results can be found in the trial open source repository (https://doi.org/10.17605/OSF.IO/U34R9).

**Secondary outcomes.** One subject (3.7%) died in-hospital in the hydroxychloroquine plus standard care group and one subject (3.9%) died in-hospital in the standard care alone group. There were no further mortalities up to day 30. There was little difference in clinical status based on the 7-point scale at day 14 after randomization (cumulative odds ratio 1.11 [95% CI 0.31 to 4.01], Supplementary Table 1). The time from randomization to hospital discharge was similar between the two groups ($p$ by log-rank test = 0.71; Supplementary Fig. 1). There was little difference in the change in NEWS2 from randomization to 96 h post-randomization (marginal mean change 0.47 [95% CI −0.58 to 1.53] points in the hydroxychloroquine group; 0.29 [95% CI −0.88 to 1.46] points in the standard care group; difference between groups 0.18 [95% CI −1.40 to 1.76] points; Supplementary Table 2 and Supplementary Fig. 2).

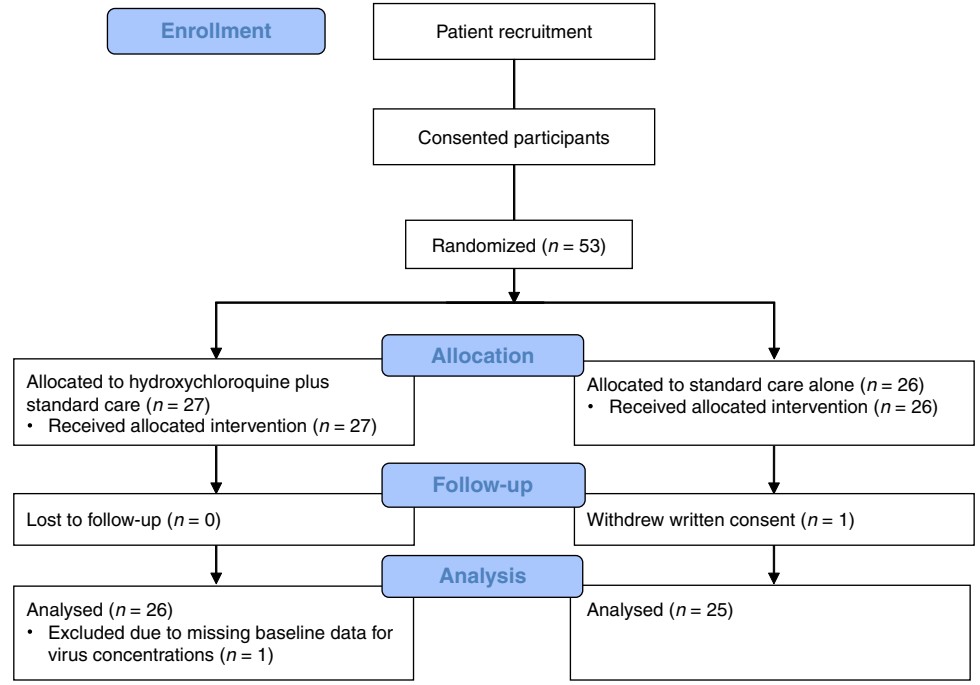

**Fig. 1 CONSORT diagram.** Flow diagram of the NO COVID-19 Study according to CONsolidated Standards Of Reporting Trials (CONSORT).

| | All (n = 53) | Hydroxychloroquine plus standard care (n = 27) | Standard care (n = 26) |
|---|---|---|---|
| **Table 1 Demographics, baseline characteristics, and safety during the trial.** | | | |
| Age (years) | 62 (50, 73) | 56 (41, 72) | 69 (51, 74) |
| Male sex, n (%) | 35 (66.0%) | 19 (70.4%) | 16 (61.5%) |
| Body mass index (kg/m²) | 26.4 (23.9, 30.5) | 25.6 (23.9, 29.4) | 27.6 (24.2, 33.0) |
| Current smoker, n (%) | 2 (3.8%) | 1 (3.7%) | 1 (3.8%) |
| Time from symptom onset to randomization (days) | 8 (7, 12) | 8 (7, 13) | 8 (6, 11) |
| *Coexisting conditions* | | | |
| Hypertension, n (%) | 17 (32.1%) | 6 (22.2%) | 11 (42.3%) |
| Diabetes mellitus, n (%) | 9 (17.0%) | 4 (14.8%) | 5 (19.2%) |
| Coronary heart disease, n (%) | 5 (9.4%) | 3 (11.1%) | 2 (7.7%) |
| Obstructive pulmonary disease, n (%) | 14 (26.4%) | 5 (18.5%) | 9 (34.6%) |
| Obesity, n (%) | 16 (30.8%) | 5 (19.2%) | 11 (42.3%) |
| ≥1 coexisting condition, n (%) | 33 (62.3%) | 14 (51.9%) | 19 (73.1%) |
| *On admission* | | | |
| Systolic blood pressure (mmHg) | 134 (124, 144) | 129 (120, 142) | 137 (130, 145) |
| Diastolic blood pressure (mmHg) | 75 (71, 85) | 75 (70, 87) | 74 (71, 79) |
| Heart rate (beats per minute) | 86 (80, 98) | 88 (76, 98) | 86 (80, 100) |
| Respiratory rate (breaths per minute) | 24 (20, 32) | 22 (20, 30) | 26 (20, 32) |
| Oxygen saturation (%) | 95 (93, 96) | 95 (94, 96) | 95 (92, 96) |
| NEWS2 | 5 (2, 6) | 4 (2, 6) | 5 (3, 7) |
| Body temperature (°C) | 38.2 (37.5, 38.7) | 38.2 (37.3, 38.7) | 38.2 (37.5, 38.6) |
| Body temperature > 37.8 °C, n (%) | 35 (66.0%) | 17 (63.0%) | 18 (69.2%) |
| Supplemental oxygen, n (%) | 20 (37.7%) | 8 (29.6%) | 12 (46.2%) |
| *Safety* | | | |
| Adverse events[a] | 237 | 125 | 112 |
| Serious adverse events, n (%)[b] | 11 (20.8%) | 5 (18.5%) | 6 (23.1%) |
| Suspected unexpected serious adverse reactions, n (%) | 1[c] (1.9%) | 0 (0.0%) | 1[c] (3.8%) |

*NEWS2* National Early Warning Score 2.
Obesity was defined as body mass index of 30 kg/m² or above. All values are presented as median with interquartile range for continuous variables or absolute numbers with percentages for categorical variables. One patient in hydroxychloroquine plus standard care had missing data for body mass index, and values are calculated based on available information.
[a]Adverse events of special interest were assessed daily and included visual disturbances, gastrointestinal discomfort, diarrhea, headache, nausea, or dizziness.
[b]No patient had more than one serious adverse event. Serious adverse events included acute respiratory distress syndrome (n = 1), pneumonia (n = 2), respiratory failure (n = 7), and urinary tract infection (n = 1).
[c]Urinary tract infection.

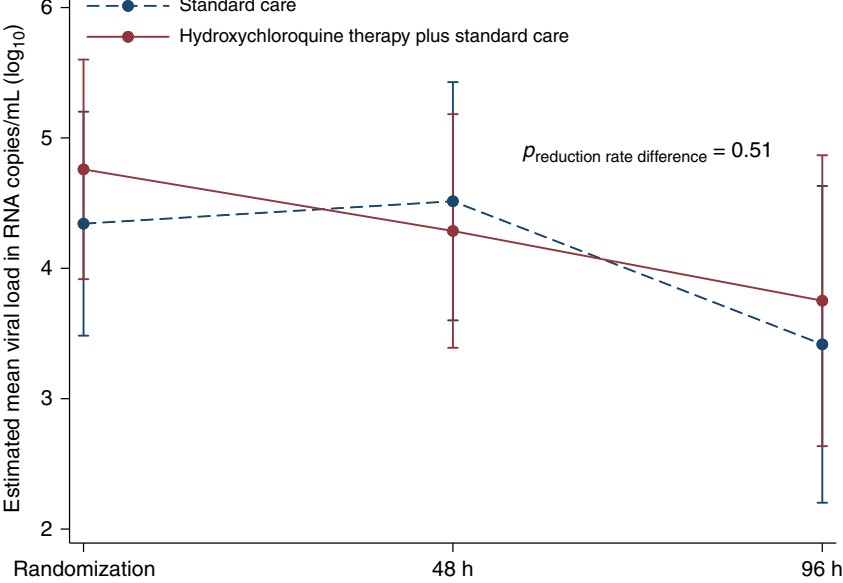

**Fig. 2 Oropharyngeal viral load (log₁₀) in hydroxychloroquine plus standard care versus standard care in the intention-to-treat population (n = 51; full analysis set).** One patient in the hydroxychloroquine plus standard care had missing baseline data for viral concentrations and one patient in standard care withdrew consent before viral load assessment at 48 h. Data are accordingly shown for 26 patients assigned to hydroxychloroquine plus standard care and 25 patients assigned to standard care. Estimated mean difference between groups was 0.27 (95% CI −0.92 to 1.47) log₁₀ RNA copies/mL at randomization, 0.06 (95% CI −1.15 to 1.26) log₁₀ RNA copies/mL at 48 h, and −0.16 (95% CI −1.67 to 1.36) log₁₀ RNA copies/mL at 96 h. Plot displays estimated marginal means (dots) and 95% confidence intervals (error bars).

## Discussion

In patients with moderately severe COVID-19 in need of hospital admission, treatment with hydroxychloroquine sulfate initiated median 8 days after first symptoms did not result in a significantly greater rate of decline of SARS-CoV-2 oropharyngeal viral load compared to standard care alone during the next 5 days. The study was stopped prematurely due to difficulties in recruiting patients and we cannot exclude a clinically important

difference in viral kinetics between the two arms due to lack of power. There is currently no consensus on what a meaningful decrease in SARS CoV-2 viral load after antiviral treatment would be. However, for influenza treatment, a 2 $\log_{10}$ fall within 48 h has been proposed as a meaningful endpoint to reach for future influenza therapies[3]. Our confidence interval of the treatment effect lies well within this limit, making a clinically important difference between treatment arms less likely. We did not observe any major effect of treatment with hydroxychloroquine on short-term mortality, degree of illness, duration of hospital admission or clinical status.

Measures of rate of decline in virus replication as primary end points to evaluate antiviral drug efficacy are crucial[3]. Rapid reductions in active viral replication may be essential to prevent tissue damage and to further clinical recovery, as well as reduce risk of viral complications and mortality. By exploring viral load as a continuous outcome, a more sensitive statistical index compared to dichotomy[4], the neutral result of hydroxy-chloroquine versus standard care in our study strongly suggests no major effect of hydroxychloroquine on the principal pathology in COVID-19. This model is supported by no apparent effect of hydroxychloroquine on SARS-CoV-2 negative conversion rate in Chinese patients with mild to moderate COVID-19[2]. The patients in the aforementioned report were however younger, afebrile and partly pretreated with antiviral drugs. The duration from onset of symptoms was additionally notably longer compared to the current investigation, which with median 8 days from start of symptoms to start of therapy closely mimics the typical clinical course of COVID-19 characterized by clinical deterioration and need for hospitalization admission 1 week after illness onset[5]. Accordingly, our trial extends the results of previous investigations to more acutely ill and febrile patients in need of hospital admission. In light of the COVID-19 pandemic, there is an unmet need of pharmacological interventions aimed at reducing morbidity and mortality. Several medical therapies have been suggested; e.g., glucocorticoids, convalescent plasma, specific antibodies, lopinavir-ritonavir, remdesivir, and hydroxy-chloroquine. Treatment with the glucocorticoid dexamethasone has shown effect on mortality in patients with COVID-19, but predominantly in patients requiring supplemental oxygen or mechanical ventilation[6]. The use of convalescent plasma received early recognition as a viable treatment option in critically ill COVID-19 patients[7], but results from randomized clinical trials are lacking. One of the most promising treatment modalities in the face of the COVID-19 pandemic is specific neutralizing antibodies, which would provide precision therapy overcoming the inherent lack of specificity provided by convalescent plasma[8]. Treatment with lopinavir–ritonavir has so far failed to demonstrate any benefit in COVID-19 beyond standard care[9]. The results for remdesivir appear more promising, but evidence is still conflicting[10,11]. Retrospective data examining the clinical effect of hydroxychloroquine in COVID-19 are also diverging[12,13], and properly conducted and adequately powered randomized trials with peer-reviewed reports are accordingly still needed to assess the therapeutic value of hydroxychloroquine on clinical outcomes in patients with COVID-19.

Our current investigation has several limitations. The study was nonblinded without placebo treatment and we recognize that the lack of blinding may have influenced the standard care treatment and decision making by the treating physician, ultimately affecting our results. However, the study outcome was SARS-CoV-2 viral load, and study personnel performing the RT-qPCR and statistical analyses were blinded concerning group allocation. We assume that change in viral load in the upper respiratory tract is a valid measure for ongoing viral replication,

but we did not perform analyses differentiating viable from nonviable virus. Oropharyngeal samples were obtained for the viral analyses, contrary to the common practice of nasopharyngeal sampling for the diagnosis of upper airway respiratory viruses. Nasopharyngeal sampling is however associated with significant discomfort for the patient, possibly to a degree leading to study discontinuation. A recent report by Wölfel et al.[14] found no significant differences in viral load when comparing naso- and oropharyngeal sampling for SARS-CoV-2. Wang et al.[10] found comparable viral loads in upper and lower respiratory tract samples, suggesting that SARS-CoV-2 viral kinetics can be studied in the upper respiratory tract. Stringent scientific support for this assumption is however still lacking. Electrocardiograms were not routinely taken during trial conduction, barring us from assessing the effect of hydroxychloroquine therapy on corrected QT interval. Finally, due to early study cessation, sample size was less than planned with resulting lower study power. Based on the exact effect estimates and standard deviations observed in this report, a future clinical trial would require a sample size of at least 928 subjects (allocation ratio 1:1 with 464 in each arm) to detect a significant difference between groups with a two-sided $\alpha = 0.05$ and $\beta = 0.80$. Sample size calculations under the assumption of increasing effect sizes of the intervention effect, and constant standard deviations from the current report are listed in Supplementary Table 3 and Supplementary Fig. 3.

In conclusion, therapy with hydroxychloroquine did not impact SARS-CoV-2 viral kinetics in patients admitted to hospital with moderately severe COVID-19. Our results suggest no important antiviral effect of hydroxychloroquine in humans infected with SARS-CoV-2.

## Methods

**Trial design**. The NO COVID-19 Study is a single center, two-arm, open label, group-sequential, pragmatic randomized controlled trial of hydroxychloroquine sulfate in adults hospitalized with COVID-19. Patients were randomly assigned to receive hydroxychloroquine sulfate (at a dose of 400 mg twice daily for 7 days) in addition to standard care or standard care alone[15]. Standard care was similar for all patients included in the study, and encompassed appropriate level and intensity of medical treatment according to local and national guidelines. No stratification was used for the computer randomization procedure. Because of rapidly decreasing incidence of COVID-19 in Norway, the trial was prematurely stopped by the trial sponsor on May 25, 2020. The study protocol was approved by the Norwegian Regional Committees for Medical Research Ethics (REC no. 121446) and the Norwegian Medicines Agency. The study was performed according to standard rules for Good Clinical Practice, with statistical methods and stopping rules described in the protocol and statistical analysis plan, with detailed descriptions in the trial open source repository (https://doi.org/10.17605/OSF.IO/U34R9).

**Patients**. All reverse transcriptase polymerase chain reaction (RT-qPCR) SARS-CoV-2 positive patients 18 years of age or older were eligible for study inclusion, if they had moderately severe COVID-19 at admission (NEWS2[16] of 6 or less). For patients who tested positive for SARS-CoV-2 before admission, SARS-CoV-2 status was verified with the external laboratory. Exclusion criteria included (1) the need of admission to intensive care unit on hospital admission, (2) history of psoriasis, (3) reduced hearing/tinnitus, (4) visual impairment, (5) known adverse reaction to hydroxychloroquine sulfate, (6) pregnancy, or (7) prolonged corrected QT interval (>450 ms). All study participants provided written informed consent before study inclusion.

Data on coexisting conditions were acquired from the hospital electronic patient records. Coronary artery disease was defined as history of myocardial infarction, coronary artery bypass grafting, or percutaneous coronary intervention. Diabetes was defined as history of diabetes mellitus type 1 or type 2, and the use of antidiabetic medication. Hypertension was defined as history of hypertension and the use of antihypertensive medication. Obstructive pulmonary disease was defined as history of chronic obstructive pulmonary disease or asthma. Obesity was defined as body mass index of 30 kg/m$^2$ or above. Current smoking was defined as daily consumption of cigarettes.

**Primary outcome**. The primary outcome was rate of decline in SARS-CoV-2 viral load in the oropharynx from baseline through the first 96 h after randomization, using a single batch of swabs and a standardized sampling procedure to saturate

them. Oropharyngeal swab samples were taken from patients at inclusion, at 48 h and at 96 h, by a selected group of study physicians. For analysis, total nucleic acids were extracted from 300 µL of each specimen using the Maxwell® RSC Viral total Nucleic Acid Purification Kit (Promega, Madison, Wisconsin, USA) according to the manufacturer's instructions and eluted in 50 µL nuclease-free water. SARS-CoV-2 detection was performed in duplicate by RT-qPCR on 5 µL nucleic acid eluate in a total reaction volume of 25 µL on a QuantStudio™ 7 Flex Real-Time PCR System (Thermofisher Scientific, Waltham, Massachusetts, USA), according to the protocol published in January 2020 by Corman et al.[17] that targets the viral E-gene of sarbecoviruses. For each patient, all samples in the time series were analyzed in the same extraction and PCR set-up. Single batches of all reagents for extraction and PCR were used for all samples in the study. SARS-CoV-2 RNA quantitation was calculated using a serial dilution of a the synthetic Wuhan coronavirus 2019 E gene RNA control comprising the viral region to be amplified, provided by the European Virus Archive Global (EVAg). Viral loads are expressed in $\log_{10}$ RNA copies/mL transport medium. The limits of detection (LoD) and quantitation (LoQ) of the assay are of 2.11 and 2.55 $\log_{10}$ RNA copies/mL, respectively. For data analyses, results below LoD (SARS-CoV-2 RNA not detected) were set to 0 $\log_{10}$ RNA copies/mL, and results below LoQ were set to the mean between LoD and LoQ values (i.e., 2.36 $\log_{10}$ RNA copies/mL). A qPCR assay targeting human β-globin was performed on all samples, where no viral RNA was detected for assessment of sample adequacy[18]. For the intention-to-treat population ($n = 51$), all actual samples were positive for human β-globin analysis, indicating adequate sample quality. Further details regarding study sampling and analysis can be found in the Supplementary Methods.

**Secondary outcomes**. The secondary outcomes include in-hospital mortality, mortality at 30 days, clinical status on a 7-point ordinal scale (1. dead, 2. hospitalized, on invasive mechanical ventilation or extracorporeal membrane oxygenation, 3. hospitalized, on noninvasive ventilation or high flow oxygen devices, 4. hospitalized, requiring supplemental oxygen, 5. hospitalized, not requiring supplemental oxygen, 6. not hospitalized, but unable to resume normal activities, 7. not hospitalized, with resumption of normal activities) at 14 days after randomization, duration of hospital admission after randomization and change in degree of illness as quantified by NEWS2[16] from randomization to 96 h. Further details on the secondary outcomes can be found in the study protocol in the trial open source repository (https://doi.org/10.17605/OSF.IO/U34R9).

**Statistical analyses**. The analysis of the primary outcome was pre-specified and detailed in the statistical analysis plan (https://doi.org/10.17605/OSF.IO/U34R9). The primary outcome was analyzed using a generalized linear mixed model, with subject-specific random intercept and slope in the full analysis set (FAS, all randomized subjects who have had at least one baseline and one postrandomization evaluation of efficacy). The analysis of the secondary endpoints was based on the prespecified description of analysis of secondary continuous, categorical and time-to-event endpoints in the study protocol (version 1.3, dated 26.03.2020). The analysis of the ordinal endpoint was not specified in the protocol and was decided post hoc before secondary endpoint analysis. The secondary endpoints were all analyzed in the FAS and based on available data. Because of the low number of deaths (one in each arm) in-hospital and at 30 days after randomization, we used descriptive statistics only for these endpoints. We used descriptive statistics to report the number of participants in each level of the 7-point clinical status scale at day 14 after randomization. We performed an ordinal logistic regression with the 7-point scale as dependent and group allocation as independent variable to compare the two study arms. We used the Kaplan–Meier method to calculate the time from randomization to hospital discharge in each study arm. Deceased participants were censored at the time of death. We used a log-rank test for equality of survivor functions to compare time from randomization to hospital discharge in the two groups. NEWS2 was assessed at several time points each day during in-hospital stay. We calculated the daily mean NEWS2 for each participant from randomization to 96 h postrandomization. We analyzed change in NEWS2 from baseline to 96 h postrandomization using a linear mixed model with fixed treatment by time and random intercept. The complete statistical analysis plan can be found in the trial open source repository (https://doi.org/10.17605/OSF.IO/U34R9). The statistical analyses were performed with STATA 16 (StataCorp LP, College Station, TX).

**Reporting summary**. Further information on research design is available in the Nature Research Reporting Summary linked to this article.

## Data availability
Anonymized data, study protocol and statistical analysis plan are available from the trial open source repository (https://doi.org/10.17605/OSF.IO/U34R9). Requests for data not included in the Manuscript, Supplementary Information or the trial open source repository should be directed to the corresponding author (H.R.).

## Code availability
The statistical code used for analysis can be found in the trial open source repository (https://doi.org/10.17605/OSF.IO/U34R9).

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

## Acknowledgements
We thank Shuo-Wang Qiao (Institute of Clinical Medicine, Faculty of Medicine, University of Oslo, Oslo, Norway) and Jan Haug Anonsen (NORCE Norwegian Research Centre AS, Bergen, Norway) for providing virus transport medium, and the Unit of Molecular Biology, Department of Multidisciplinary Laboratory Medicine and Medical Biochemistry, Division for Diagnostics and Technology, Akershus University Hospital, Lørenskog, Norway, for pre-study optimization of viral measurements. The publication was supported by the European Virus Archive Global (EVA-GLOBAL) project that has received funding from the European Union's Horizon 2020 research and innovation programme under grant agreement No 871029. We thank Haldor Husby and the Unit of Data Analysis at the Division of Research and Innovation, Akershus University Hospital, Lørenskog, Norway, for valuable help with clinical data acquisition from the data warehouse at Akershus University Hospital, Lørenskog, Norway. We thank Randi Kristoffersen and Lisbeth Johnsen (Clinical Trial Unit, Akershus University Hospital, Lørenskog, Norway) for help with trial planning, management and regulatory approvals. We thank Jannicke Dokken, Merete Moen Tollefsen, Jessica Andreassen and Ingunn Melkeraaen (Department of Infectious Diseases, Division of Medicine, Akershus University Hospital, Lørenskog, Norway) for patient inclusion and follow-up. We thank all collaborators at the Clinical Trial Unit, Oslo University Hospital, Oslo, Norway, for

valuable help with trial management, data handling and monitoring, and preparation of the electronic case report form. We thank all participating physicians at the Akershus University Hospital, Lørenskog, Norway, for help with patient screening and inclusion. Most importantly, we thank all the patients who participated in the trial.

## Author contributions

M.N.L., coordinating investigator, prepared the trial protocol, handled regulatory approvals, drafted the manuscript. J.E.B., investigator, study sponsor, critically reviewed the manuscript and approved the final manuscript as submitted. A.E., investigator, inclusion of patients, critically reviewed the manuscript and approved the final manuscript as submitted. D.K., investigator, laboratory and analytical support, critically reviewed the manuscript and approved the final manuscript as submitted. I.C.O., trial statistician, prepared the trial protocol, critically reviewed the manuscript, and approved the final manuscript as submitted. C.S.R., trial statistician, critically reviewed the manuscript and approved the final manuscript as submitted. A.R., investigator, laboratory and analytical support, critically reviewed the manuscript and approved the final manuscript as submitted. C.M.J., investigator, laboratory and analytical support, critically reviewed the manuscript and approved the final manuscript as submitted. T.O., investigator, critically reviewed the manuscript and approved the final manuscript as submitted. H.R., investigator, conceived the idea for the study, prepared the trial protocol, handled regulatory approvals, critically reviewed the manuscript, and approved the final manuscript as submitted. O.D., principal investigator, prepared the trial protocol, handled regulatory approvals, inclusion of patients, critically reviewed the manuscript, and approved the final manuscript as submitted.

## Competing interests

The authors declare no competing interests.
