## [Peer Review File · Nature Communications]

REVIEWERS' COMMENTS

Reviewer #1 (Remarks to the Author):

The authors randomized 53 adult patients hospitalized with COVID-19 to hydroxychloroquine therapy (400 mg twice daily) vs. standard care alone. The treatment did not influence the rate of decline of viral load during the first five days.

These observations indicate (or confirm) that treatment with hydroxychloroquine has no significant antiviral effect.

We no longer believe much in the efficacy of hydroxychloroquine, but this can be an additional nail in the coffin.

The study is well performed and the data are credible. The paper is well written.

Minor comment

-'Three medical therapies were early 90 in the COVID-19 pandemic considered to be strong candidates...': I would not limit it to three interventions. In my mind, the most promising therapy is the use of specific antibodies, but there are several other options considered.

JL Vincent

Reviewer #2 (Remarks to the Author):

This is a well written randomized clinical trial testing hydroxychloroquine in hospitalized patients with COVID-19.

This manuscript is a well written, and there are limited comments -- other than the largest problem that this is a n=53 trial that was halted early -- once COVID-19 cases abated in Norway. Still, this manuscript provides useful information in the grand context.

The small sample size does limit the conclusions that can be drawn from the manuscript; however, the use of a continuous variable for the primary endpoint was a wise choice. Ultimately, the trial is a successful in demonstrating that there is not a substantial reduction in oropharyngeal viral load.

A useful contribution for the literature would be to provide more information for future studies for estimated power calculations and sample sizes. Instead of just saying the trial had less statistical power, it would be more helpful to specify what sample size would be needed based on the standard deviation observed. Providing the standard deviation and several possible effect sizes based on samples sizes may be helpful for others in designing future trials, starting with $n=25$ and moving upwards. This might be the most helpful aspect of this manuscript.

Reviewer #3 (Remarks to the Author):

This was a well designed group sequential adaptive trial planning to recruit up to 202 patients with moderate COVID, with three interim analyses which could allow for early stopping. The trial was stopped early due to poor recruitment and therefore only the single interim assessment had been undertaken and the results are presented for 51 patients (53 were recruited and randomised, but one in each group was withdrawn). There were appropriate adjustments to the alpha in order to control type 1 error. The statistical analysis plan overall is well developed and appropriate for an adaptive drug trial.

There are a few things in the reporting and analysis however which reduce the value/quality of this report and its conclusions. I think it is important that the main methods in the report acknowledge the design and impact of stopping early. In order to conclude that there is no effect in such an underpowered study it is important that the clinically important difference is discussed, rather than the lack of a significant difference between groups. The actual rates of reduction found in the study are so different to those assumed in the sample size calculation (6 and 8 in the calculation and 0.14 and 0.24 in the study), that I kept trying to see if they were measured in different units, but it did not appear to be the case. This is where the discussion needs to focus - is there a potentially clinically useful reduction in viral load that has not been found significant? What is presented does not justify the no difference conclusion as it stands (but this may be justified by better presentation and discussion). The early stopping plans seemed to only be for superiority and not futility, so it is important to discuss further.

As this is a pragmatic study it is vital that what happened in standard care is described in both arms, in order to understand if A+B is better than B we must know what B was (and if it was the same in both arms). It also seems strange that there is a SUSAR in the standard care arm, when these are defined as only occurring in the IMP arm.

As this is the main report of this trial I would expect to see all secondary outcomes being presented, even if only in the supplementary material.

A minor point is that it is not clear if the randomisation was stratified in any way and if so what variables were used.

Kerry Hood

A pragmatic randomized controlled trial reports lack of efficacy of hydroxychloroquine on coronavirus disease 2019 viral kinetics

Reviewer #1 (Remarks to the Author):

The authors randomized 53 adult patients hospitalized with COVID-19 to hydroxychloroquine therapy (400 mg twice daily) vs. standard care alone. The treatment did not influence the rate of decline of viral load during the first five days. These observations indicate (or confirm) that treatment with hydroxychloroquine has no significant antiviral effect. We no longer believe much in the efficacy of hydroxychloroquine, but this can be an additional nail in the coffin. The study is well performed and the data are credible. The paper is well written.

Minor comment

-‘Three medical therapies were early 90 in the COVID-19 pandemic considered to be strong candidates...’: I would not limit it to three interventions. In my mind, the most promising therapy is the use of specific antibodies, but there are several other options considered.

We thank the Reviewer for the evaluation and comments that have improved the quality of our manuscript. We agree with the Reviewer that there are indeed several promising therapeutics in COVID-19. In the revised version of our manuscript, we have accordingly added the following to the Discussion (page 7, line 1):

“In light of the COVID-19 pandemic, there is an unmet need of pharmacological interventions aimed at reducing morbidity and mortality. Several medical therapies have been suggested; e.g. glucocorticoids, convalescent plasma, specific antibodies, lopinavir-ritonavir, remdesivir, and hydroxychloroquine. Treatment with the glucocorticoid dexamethasone has shown effect on mortality in patients with COVID-19, but predominantly in patients requiring supplemental oxygen or mechanical ventilation. The use of convalescent plasma received early recognition as a viable treatment option in critically ill COVID-19 patients, but results from randomized clinical trials are lacking. One of the most promising treatment modalities in the face of the COVID-19 pandemic is specific neutralizing antibodies, which would provide precision therapy overcoming the inherent lack of specificity provided by convalescent plasma.”

JL Vincent

Reviewer #2 (Remarks to the Author):

This is a well written randomized clinical trial testing hydroxychloroquine in hospitalized patients with COVID-19. This manuscript is a well written, and there are limited comments -- other than the largest problem that this is a n=53 trial that was halted early -- once COVID-19 cases abated in Norway. Still, this manuscript provides useful information in the grand context.

The small sample size does limit the conclusions that can be drawn from the manuscript; however, the use of a continuous variable for the primary endpoint was a wise choice. Ultimately, the trial is a successful in demonstrating that there is not a substantial reduction in oropharyngeal viral load.

A useful contribution for the literature would be to provide more information for future studies for estimated power calculations and sample sizes. Instead of just saying the trial had less statistical power, it would be more helpful to specify what sample size would be needed based on the standard deviation observed. Providing the standard deviation and several possible effect sizes based on samples sizes may be helpful for others in designing future trials, starting with n=25 and moving upwards. This might be the most helpful aspect of this manuscript.

We thank the Reviewer for a thorough evaluation of our manuscript and for important and helpful comments. We agree that an elaboration on sample sizes for possible future studies would be of help to the readers of our manuscript. Accordingly, we have added the following passage to the Discussion (page 8, line 7), with further details in Supplementary Table 2 and Supplementary Fig. 3:

*“Based on the exact effect estimates and standard deviations observed in this report, a future clinical trial would require a sample size of at least 928 subjects (allocation ratio 1:1 with 464 in each arm) to detect a significant difference between groups with a two-sided $\alpha = 0.05$ and $\beta = 0.80$. Sample size calculations under the assumption of increasing effect sizes of the intervention effect and constant standard deviations from the current report are listed in **Supplementary Table 3** and **Supplementary Fig. 3.**”*

Reviewer #3 (Remarks to the Author):

This was a well designed group sequential adaptive trial planning to recruit up to 202 patients with moderate COVID, with three interim analyses which could allow for early stopping. The trial was stopped early due to poor recruitment and therefore only the single interim assessment had been undertaken and the results are presented for 51 patients (53 were recruited and randomised, but one in each group was withdrawn). There were appropriate adjustments to the alpha in order to control type 1 error. The statistical analysis plan overall is well developed and appropriate for an adaptive drug trial.

There are a few things in the reporting and analysis however which reduce the value/quality of this report and its conclusions. I think it is important that the main methods in the report acknowledge the design and impact of stopping early. In order to conclude that there is no effect in such an underpowered study it is important that the clinically important difference is discussed, rather than the lack of a significant difference between groups. The actual rates of reduction found in the study are so different to those assumed in the sample size calculation (6 and 8 in the calculation and 0.14 and 0.24 in the study), that I kept trying to see if they were measured in different units, but it did not appear to be the case. This is where the discussion needs to focus - is there a potentially clinically useful reduction in viral load that has not been found significant? What is presented does not justify the no difference conclusion as it stands (but this may be justified by better presentation and discussion).

We thank the Reviewer for the evaluation and comments that have improved the quality of our manuscript. The literature on changes in viral kinetics in acute viral infections, both the natural course and how this may be influenced by medical interventions, is scarce, making sample size calculations in this kind of clinical trial challenging. We agree with the Reviewer that clinically important differences may be present despite lack of statistically significant differences between intervention groups. In a report by Ison et al. (Ison et al. End points for testing influenza antiviral treatments for patients at high risk of severe and life-threatening disease. *The Journal of infectious diseases* 2010;201:1654-62.), a 2 log₁₀ fall within 48 hours has been proposed as a meaningful endpoint to reach in studies on influenza treatment, a clinical situation comparable to the current investigation. To further clarify this lack of both clinical and statistical difference, we have added the following to the Discussion of the revised manuscript (page 6, line 5):

*“The study was stopped prematurely due to difficulties in recruiting patients and we cannot exclude a clinically important difference in viral kinetics between the two arms due to lack of power. There is currently no consensus on what a meaningful decrease in SARS CoV-2 viral load after antiviral treatment would be. However, for influenza treatment, a 2 log₁₀ fall within 48 hours has been proposed as a meaningful endpoint to reach for future influenza therapies. (Ison et al. End points for testing influenza antiviral treatments for patients at high risk of severe and life-threatening disease. *The Journal of infectious diseases* 2010;201:1654-62.) Our confidence interval of the treatment effect lies well within this limit, making a clinically important difference between treatment arms less likely. We did not observe any major effect of treatment with hydroxychloroquine on short-term mortality, degree of illness, duration of hospital admission or clinical status.”*

The early stopping plans seemed to only be for superiority and not futility, so it is important to discuss further.

We appreciate this comment by the Reviewer regarding early stopping of the study. The reason we did not plan to stop for futility was that at the planning stage of the study, during the initial phase of the pandemic, we anticipated a large inclusion rate and fulfillment of the sample size within a short

time. In addition, as COVID-19 was a new illness, we did not have exact estimates of what a clinically significant effect would be. Hence, stopping for futility with a conclusion of no clinically significant difference would be difficult and possibly lead to an erroneous result of non-conclusive stopping, which we considered a suboptimal situation.

As this is a pragmatic study it is vital that what happened in standard care is described in both arms, in order to understand if A+B is better than B we must know what B was (and if it was the same in both arms).

We agree that standard care should be detailed further. In the revised version of the manuscript, we have added the following passage to the Methods (page 8, line 21):

“Standard care was similar for all patients included in the study, and encompassed appropriate level and intensity of medical treatment according to local and national guidelines.”

It also seems strange that there is a SUSAR in the standard care arm, when these are defined as only occurring in the IMP arm.

We agree with the Reviewer that reporting of SUSAR for both intervention arms may seem unorthodox. For the sake of transparency and to correctly address the proportion of events, we did however report all events that occurred during trial conduction, for both standard care and standard care plus IMP.

As this is the main report of this trial I would expect to see all secondary outcomes being presented, even if only in the supplementary material.

We agree with the Reviewer on this point, and in the revised version of our manuscript we have included results of the secondary outcomes of the trial. Currently, we do not have data from the biochemical analyses or the 90 day mortality data, barring us from presenting results from these secondary outcomes. All the other secondary outcome are now reported in the Results of the revised manuscript (page 5, line 16):

*“One subject (3.7%) died in-hospital in the hydroxychloroquine group plus standard care group and one subject (3.9%) died in-hospital in the standard care alone group. There were no further mortalities up to day 30. There was little difference in clinical status based on the 7 point scale at day 14 after randomization (cumulative odds ratio 1.11 [95% CI 0.31 to 4.01], **Supplementary Table 1**). The time from randomization to hospital discharge was similar between the two groups (p by log-rank test = 0.71; **Supplementary Fig. 1**). There was little difference in the change in NEWS2 from randomization to 96 hours post-randomization (marginal mean change 0.47 [95% CI -0.58 to 1.53] points in the hydroxychloroquine group; 0.29 [95% CI -0.88 to 1.46] points in the standard care group; difference between groups 0.18 [95% CI -1.40 to 1.76] points; **Supplementary Table 2** and **Supplementary Fig. 2**).“*

We further expanded the methods section to describe these additional analyses (page 10, line 21):

“Secondary outcomes

The secondary outcomes include in-hospital mortality, mortality at 30 days, clinical status on a 7-point ordinal scale (1, dead; 2, hospitalized, on invasive mechanical ventilation or extracorporeal membrane oxygenation; 3, hospitalized, on non-invasive ventilation or high flow oxygen devices; 4, hospitalized, requiring supplemental oxygen; 5, hospitalized, not requiring supplemental oxygen; 6, not hospitalized, but unable to resume normal activities; 7, not hospitalized, with resumption of normal activities) at 14 days after randomization, duration of hospital admission after randomization

and change in degree of illness as quantified by NEWS216 from randomization to 96 hours. Further details on the secondary outcomes can be found in the study protocol in the trial open source repository (<https://doi.org/10.17605/OSF.IO/U34R9>).

Statistical analyses

The analysis of the primary outcome was pre-specified and detailed in the statistical analysis plan (<https://doi.org/10.17605/OSF.IO/U34R9>). The primary outcome was analyzed using a generalized linear mixed model, with subject-specific random intercept and slope in the full analysis set (FAS, all randomised subjects who have had at least one baseline and one post-randomisation evaluation of efficacy). The analysis of the secondary endpoints was based on the pre-specified description of analysis of secondary continuous, categorical and time-to-event endpoints in the study protocol (version 1.3, dated 26.03.2020). The analysis of the ordinal endpoint was not specified in the protocol and was decided post hoc before secondary endpoint analysis. The secondary endpoints were all analyzed in the FAS and based on available data. Because of the low number of deaths (one in each arm) in-hospital and at 30 days after randomization, we used descriptive statistics only for these endpoints. We used descriptive statistics to report the number of participants in each level of the 7-point clinical status scale at day 14 after randomization. We performed an ordinal logistic regression with the 7-point scale as dependent and group allocation as independent variable to compare the two study arms. We used the Kaplan-Meier method to calculate the time from randomization to hospital discharge in each study arm. Deceased participants were censored at the time of death. We used a log-rank test for equality of survivor functions to compare time from randomization to hospital discharge in the two groups. NEWS2 was assessed at several time points each day during in-hospital stay. We calculated the daily mean NEWS2 for each participant from randomization to 96 hours post-randomization. We analyzed change in NEWS2 from baseline to 96 hours post-randomisation using a linear mixed model with fixed treatment by time and random intercept. The complete statistical analysis plan can be found in the trial open source repository (<https://doi.org/10.17605/OSF.IO/U34R9>). The statistical analyses were performed with STATA 16 (StataCorp LP, College Station, TX)."

A minor point is that it is not clear if the randomisation was stratified in any way and if so what variables were used.

We agree with the Reviewer that the randomization procedure requires further detail. In the revised version of the manuscript, we have changed the following passage in the Methods accordingly (page 8, line 20):

"Patients were randomly assigned to receive hydroxychloroquine sulphate (at a dose of 400 mg twice daily for seven days) in addition to standard care or standard care alone. No stratification was used for the computer randomization procedure."

Kerry Hood